



# urbanChemFoam 1.0: Large-Eddy Simulation of Non-Stationary Chemical Transport of Traffic Emissions in an Idealized Street Canyon

Edward C. Chan[1] and Timothy M. Butler[1]

[1]Institute for Advanced Sustainability Studies (IASS), Potsdam, Germany

**Correspondence:** E.C. Chan (edward.chan@iass-potsdam.de)

**Abstract.** This paper describes a large-eddy simulation based chemical transport model, developed under the OpenFOAM framework, implemented to simulate dispersion and chemical transformation of nitrogen oxides from traffic sources in an idealized street canyon. The dynamics of the model, in terms of mean velocity and turbulent fluctuation, are evaluated using available stationary measurements. A transient model run using a photostationary reaction mechanism for nitrogen oxides and ozone subsequently follows, where non-stationary conditions for meteorology, background concentrations, and traffic emissions are applied over a 24-hour period, using regional model data and measurements obtained for the City of Berlin in July, 2014. Diurnal variations of pollutant concentrations indicate dependence on emission levels, background concentrations, and solar state. Comparison of vertical and horizontal profiles with corresponding stationary model runs at select times show that, while there are only slight differences in velocity magnitude, visible changes in primary and secondary flow structures can be observed. In addition, temporal variations in diurnal profile and cumulative species concentration result in significant deviations in computed pollutant concentrations between transient and stationary model runs.

**Keywords.** Urban scale; Street Canyon; Large-Eddy Simulation (LES); Chemical Transport Model (CTM); Emissions

## 1 Introduction

The study of dispersion and chemical transformation of pollutants at urban scales (less than 200 m; Britter and Hanna, 2003) is not only of immediate scientific and engineering interest, it is also an indispensable exercise for socio-economists and policy makers seeking to quantify their decisions. It is at this scale where the local population is in close proximity with the pollutant sources, such as nitrogen oxides ($NO_X$) from traffic sources. Based on recent reports from the European Environmental Agency, nitric oxide (NO) and nitrogen dioxde $NO_2$ derived from road traffic constitute close to 40 % of total $NO_X$ emissions in the EU-28 nations (EEA, 2019a). In addition to being an irritant and a potential carcinogen, $NO_2$ acts as precursor to a number of harmful substances, such as nitric acid ($HNO_3$), ground level ozone ($O_3$), and particulate matters (PM). Further, it is estimated that the associated aggregated global economic impact, including diminished productivity, health care expenditures, and reduced agricultural crop yields, could increase from from 330 billion USD in 2015 to up to 3300 billion USD in 2060 (OECD, 2016).





In 2017, about 10% of all air quality monitoring stations among the EU-28 have reported $NO_X$ exceedances of an annual

mean limit of 40 $\mu g \cdot m^{-3}$, with the majority observed in urban traffic stations (EEA, 2019b). In Germany, $NO_X$ exceedances were reported in over 40% of urban traffic stations in 2018 (Minkos et al., 2020). Although these figures already represent a continually downward trend from 2014 levels (EEA, 2019a), additional measures are still required to meet the European air quality targets (EEA, 2019b). In conjunction with rigorous air quality monitoring programs, and regional-scale modelling (Kuik et al., 2016; von Schneidemesser et al., 2017), urban-scale modelling can offer additional insights on localized formation

and dispersion of pollutants and their impact on air quality at regional and urban scales, thus providing actors in environmental policy crucial information in identifying effective mitigation strategies.

As basic geometrical unit in a typical urban environment, a street canyon consists of two rows of buildings separated by a road section. The aspect ratio between the building height ($H$) and the road section width ($W$) of the canyon is an important geometrical characteristic, as it has significant influence on pollutant transport (Liu et al., 2005). The flow in the freestream

region above the canyon induces recirculation zones within the canyon, as demonstrated in numerical studies of infinite canyon arrangements (Baik and Kim, 1998; Li et al., 2008), which restricts exchange of air mass to the shear layer between the canyon and freestream regions (Driver and Seegmiller, 1985). The geometrical arrangement of these flow structures could be altered significantly, however, when various parts of the street canyon (that is, the building walls and street surface) are maintained at different temperatures, through solar heating, for instance, which, in combination with the prevailing wind, could increase

stratification of pollutant concentrations in the canyon (Kim and Baik, 2001; Madalozzo et al., 2014; Xie et al., 2007).

The Reynolds-Averaged Navier-Stokes (RANS) equations is a popular numerical framework for modelling turbulent dynamics due to its simplicity and correspondingly low computational cost. While the RANS approach has been shown to adequately represent mean flow behaviors in some cases (Baik and Kim, 1998; Kwak and Baik, 2014), in the presence of large-scale unsteady mixing, such as vortex shedding from building surfaces, mean dynamics is not always sufficient, as these motions

are not universal according to the Kolmogorov hypothesis (Pope, 2000), as seen in part by large discrepancies observed in simulated pollutant concentrations for dispersion scenarios in comparison with measurement data (Tominaga and Stathopoulos, 2009). On the other hand, using large-eddy simulation (LES), large-scale turbulent motions can be resolved explicitly, leaving closure modelling to the smaller scales, which exhibit universal character to some extent. Furthermore, unlike RANS, where the prognostic equations are already averaged in time, the resolved scale statistical quantities in LES are evaluated after

the computation has taken place. Hence LES can arguably provide a much more accurate and complete representation of the evolution of turbulent flow structures, as well as chemical transport of pollutant species.

Further, a number of studies have considered the coupling of dynamics with reaction kinetics relevant to urban traffic emissions. One such simple mechanism is the photostationary $NO$ - $NO_2$ - $O_3$ mechanism (Leighton, 1961). In a street canyon, photochemical steady state is well maintained in the individual vortices as well as ground level corners where pavements are

located, while deviations are observed close to the traffic emission sources, and in the roof-level shear layer (Baker et al., 2004). Deviations from the photo steady state indicates heterogeneity due to imperfect mixing of $NO_X$ (Zhong et al., 2015), or presence of $NO_X$ oxidizing agents other than $O_3$, such as hydroperoxy ($HO_2$) or organic peroxy ($R$-$O_2$) radicals, formed mainly through oxidation of carbon monoxide ($CO$) and hydrocarbons ($R$-$H$) initiated by hydroxyl ($OH$) radicals (Carpenter et al.,





1998). The photostationary mechanism has been shown to be adequate in modelling the concentrations of $NO_X$ and $O_3$. In

comparison with more detailed mechanisms, such as the Carbon Bond Mechanism (CBM-IV), Garmory et al. (2009) reported

a difference of about 1 ppb for concentrations of NO, $NO_2$, and $O_3$. Also, in Bright et al. (2013), similar levels of discrepancy

are also observed with the Reduced Chemical Scheme (RCS).

The present study seeks to extend the current corpus on idealized street canyon modelling modelling in two aspects. First,

previous studies have been conducted in stationary meteorological and pollutant conditions, inasmuch transient studies are

carried out under statistically steady wind and constant emission rates (Kwak and Baik, 2014; Kristóf et al., 2020). Thus

the possibility of introducing diurnal variations on all forcings and their effects on the formation and spatial distribution of

pollutant species can be contemplated in this investigation. This could, for instance, help conceptualize more comprehensive

parametrization schemes, in which effects of transient conditions could be considered. Second, using the idealized street canyon

as a starting point, the chemical transport model can be further conceived and developed to accommodate more realistic

geometries, such as a city block, coupled with meteorological conditions and pollutant concentrations extracted from regional

models or taken from measurements, which also requires more detailed chemical kinetics.

## 2   Model Description

As illustrated in Figure 1, the solution domain representing an infinite deep canyon, will be used for this study throughout. To

facilitate discussion, $x$ will be referred to as the *spanwise* direction, $y$ the *lengthwise* direction, and $z$ the *vertical* direction. The

origin of the domain is located on the street level, at the intersection between the spanwise ($x-z$) and lengthwise ($y-z$) planes

of symmetry. The span of the street section ($W$) and 18 m, and the building height ($H$) is 36 m, resulting in an height-to-span

aspect ratio of two. The freestream boundary is located at 76 m above the canyon roof, or 112 m above street level, from

which meteorological conditions are specified. The canyon roof extends 9 m on each side of the spanwise direction, as shown

in Figure 1(A). The street section length is set to 40 m, as shown in Figure 1(B), centered about the domain origin. Traffic

emissions are introduced as two 1.5 m wide sections in equidistant arrangement, as shown in Figure 1(C), and are assumed

to be continuous and spatially uniform along both. Cyclic conditions are imposed on the spanwise and lengthwise boundary

pairs to ascertain periodicity in the corresponding directions, as well as to allow turbulent flow structures to freely develop

across the solution domain. To prevent accumulation of pollutants through the spanwise cyclic boundary while maintaining

background concentrations, chemical species will be introduced from the upstream spanwise boundary at specific, spatially

uniform mixing ratios, while a zero gradient boundary condition will be imposed on the downstream spanwise boundary,

allowing excess pollutants to leave the solution domain.

Initially, the dynamics of this model framework will be evaluated using the stationary laser doppler anemometry (LDA)

measurements of Li et al. (2008). This is followed by a transient simulation run with reaction kinetics activated. Boundary

conditions representative of a working day for the City of Berlin in July 2014 will be used. Diurnal variations for ambient wind

and thermodynamics are extracted on an hourly basis from the region model results of Kuik et al. (2018) using the Weather

Research and Forecasting (WRF) Model (Skamarock et al., 2008). Meanwhile, hourly traffic emissions are derived from





speciated spatio-temporal redistribution of annual $NO_X$ emission levels provided by the Berlin City Senate. Urban background concentrations for NO, $NO_2$, and $O_3$ are obtained from the Berlin Air Quality Measurement Network (*Berliner Luftgüte-Messnetz*, BLUME). In addition to diurnal variations of species concentration at specific locations in the canyon, in-canyon

vertical and horizontal profiles will be compared against corresponding stationary model runs at specific times of the day.

### 2.1   Prognostic Equations

The prognostic equations employed in the present model will be formulated using a weakly compressible framework. This generalizes the theoretical foundation, and opens up possibilities of direct coupling with other compressible regional models such as WRF in the future. These equations form the dynamic core of the modelling framework, to which the conservation

equations for mass, momentum, energy, and species to be solved numerically to determine the spatial and temporal distribution and evolution of the prognostic variables, namely, wind velocity components ($u_i$), pressure ($p$), internal energy ($e$), and mass mixing ratios for each gas-phase component species ($y_q$):

$$D_t \rho = 0 \tag{1}$$

$$D_t \left( \rho u_i \right) = -\partial_i p + \partial_j \tau_{ij} - \epsilon_{ijk} \left[ 2 \sin \left( \phi \right) \omega_j \right] \left( \rho u_k \right) \tag{2}$$

$$D_t \left[ \rho h + \frac{1}{2} \left( \rho u_i u_i \right) \right] - \partial_t p = \partial_j \left[ \partial_j e + \tau_{ij} u_i \right] \tag{3}$$

$$D_t \left( \rho y_q \right) = \partial_j \left( D_{pq} \partial_j y_q \right) + \rho r_q, \tag{4}$$

where the notation $D_t \left( \cdot \right) \equiv \partial_t \left( \cdot \right) + \partial_j u_j \left( \cdot \right)$ represents the substantial derivative, $\rho$ is the density of the gas mixture, $h \equiv e - p/\rho$ is the mixture enthalpy, and $r$ is the net chemical production rate of each gas-phase species. The Coriolis coefficient is denoted by $2 \sin \left( \phi \right) \omega_j$, as a function of angular velocity $\omega$, and the latitude of the point of interest $\phi$. The shear stress tensor $\tau_{ij}$

in Equation (2) is represented by $\tau_{ij} \equiv \mu \left[ \left( \partial_i u_j + \partial_j u_i \right) - \left( 2/3 \right) \left( \partial_j u_i \right) \delta_{ij} \right]$, where $\mu$ is the dynamic viscosity, and $\delta_{ij}$ is the Kronecker delta. Enthalpy changes from chemical reactions are deemed negligible and thus are not considered in Equation (3).

Due to the computational effort required to directly resolve all possible turbulent scales in any flow of practical interest, some form of statistical treatment must be adopted. In the LES model approach, this is realized by decomposing the instantaneous scalar variable $\varphi$ into a *resolved* scale, $\langle \varphi \rangle$, and a residual scale, $\varphi_\Delta$, through spatial convolution with a low-pass filter kernel

$F$ (Leonard, 1974):

$$\langle \varphi \rangle = \int_\forall \varphi(\xi) F( \underset{\rightarrow}{x} - \underset{\rightarrow}{\xi} \,|\, \Delta \, ) d\xi. \tag{5}$$

The filter width $\Delta$ is associated with the smallest turbulent scale that can be retained in the resolved scale. In practice, $\Delta$ is derived from the local mesh dimensions, for example, the cube root of the cell volume, such that $\Delta \equiv \left( \delta x_1 \cdot \delta x_2 \cdot \delta x_3 \right)^{1/3}$. Thus all turbulent scales residing within the resolved scale, that is, those with spatial scales larger than the attenuation threshold

defined by $F \left( \Delta \right)$, can be captured directly with the computational grid, while the residual turbulent scales, which are commonly referred to as the *subgrid* scale (SGS), are modelled.





As with Reynolds averaging for time-averaged scalars, additional terms will result from the filtering operation of equations (1 - 4), deriving from the advective term in the substantial derivative $D_t \langle \varphi \rangle$:

$$\partial_j \left( u_j \varphi \right) = \partial_j [ \, \langle u_j \rangle \langle \varphi \rangle +$$
$$\underbrace{\langle u_j \rangle \left( \varphi \right)_\Delta + \left( u_j \right)_\Delta \langle \varphi \rangle + \left( u_j \right)_\Delta \left( \varphi \right)_\Delta}_{\sigma_{ij}^\varphi} ],$$

which implies

$$\sigma_{ij}^\varphi = \langle u_j \varphi \rangle - \langle u_j \rangle \langle \varphi \rangle, \tag{6}$$

where $\sigma_{ij}^\varphi$ is known as the SGS term, and in the case of the momentum equation (2) this is referred to as the SGS stress tensor, and is denoted simply as $\sigma_{ij}$. Further, in a compressible flow framework, where density also varies, additional SGS terms from density filtering will be introduced in Equation (1). This can be avoided, however, by performing the filtering operation on a

density averaged scalar, where $\widetilde{\varphi} = \langle \rho \varphi \rangle / \langle \varphi \rangle$, through a process known as Favre filtering (Favre, 1965). The corresponding filtered transport equations for equations (1 - 4) then become:

$$\widetilde{D}_t \langle \rho \rangle = 0 \tag{7}$$
$$\widetilde{D}_t \left( \langle \rho \rangle \widetilde{u}_i \right) = -\partial_i \langle p \rangle + \partial_j \left( \widetilde{\tau}_{ij} - \widetilde{\sigma}_{ij} \right) -$$
$$\epsilon_{ijk} \left[ 2 \sin \left( \phi \right) \omega_j \right] \left( \langle \rho \rangle \widetilde{u}_k \right) \tag{8}$$

$$\widetilde{D}_t \left[ \langle \rho \rangle \widetilde{h} + \frac{1}{2} \left( \langle \rho \rangle \widetilde{u}_i \widetilde{u}_i \right) \right] - \partial_t \langle p \rangle =$$
$$\partial_j \left[ \left( \partial_j \widetilde{e} + \widetilde{\tau}_{ij} \widetilde{u}_i \right) - \left( \sigma_{ij}^h + \sigma_{ij}^{u_i} \right) \right] \tag{9}$$
$$\widetilde{D}_t \left( \langle \rho \rangle \widetilde{y}_q \right) = \partial_j \left( D_{pq} \partial_j \widetilde{y}_q - \sigma_{ij}^{y_q} \right) + \langle \rho \rangle r_q, \tag{10}$$

where the notation for Favre-filtered substatial derivative is written as $\widetilde{D}_t \left( \cdot \right) \equiv \partial_t \left( \cdot \right) + \partial_j \widetilde{u}_j \left( \cdot \right)$. In addition, Vreman et al. (1995) show that nonlinearities arising from filtering of diffusion terms can be neglected for weakly compressible flows based

on *a priori* direct numerical simulation (DNS) data. The expanded form for the corresponding SGS terms for equations (7 - 10) are presented below:

$$\widetilde{\sigma}_{ij} = \langle \rho \rangle \left( \widetilde{\widetilde{u}_i \widetilde{u}_j} - \widetilde{u}_i \widetilde{u}_j \right) \tag{11}$$
$$\widetilde{\sigma}_{ij}^h = \langle \rho \rangle \left( \widetilde{\widetilde{u}_j e} - \widetilde{u}_j \widetilde{e} \right) + \left( \widetilde{\langle p \rangle \tau_{ij}} - \langle p \rangle \widetilde{\tau}_{ij} \right) \delta_{ij} \tag{12}$$
$$\widetilde{\sigma}_{ij}^{u_i} = \frac{1}{2} \langle \rho \rangle \left( \widetilde{\widetilde{u}_i \widetilde{u}_i \widetilde{u}_j} - \widetilde{u}_i \widetilde{u}_i \widetilde{u}_j \right) \tag{13}$$

$$\widetilde{\sigma}_{ij}^{y_q} = \langle \rho \rangle \left( \widetilde{\widetilde{u}_j y_q} - \widetilde{u}_j \widetilde{y}_q \right). \tag{14}$$

Detailed discussions pertaining to the modelling and parameterization of the remaining SGS terms described in equations (12 - 14) can be found in Martín et al. (2000).





Numerous approaches are available for modelling the SGS terms (Garnier et al., 2009). A common method, first proposed by Smagorinsky (1963), is to model the SGS stresses by invoking the gradient transport theorem:

$$\widetilde{\sigma}_{ij} \approx -2\mu_\Delta \left( \widetilde{S}_{ij} - \frac{1}{3} \widetilde{S}_{ij}\delta_{ij} \right), \tag{15}$$

where $\widetilde{S}_{ij} \equiv (1/2)\left(\partial_i \widetilde{u}_j + \partial_j \widetilde{u}_i\right)$. The closure of subgrid scale viscosity $\mu_\Delta$ is, in turn, achieved by using representative turbulent scales. Based on the one equation model of Deardorff (1980) for incompressible flows, Yoshizawa (1986) proposed a similar closure scheme for weakly compressible flows using the subgrid scale turbulent kinetic energy (TKE) $k_\Delta \equiv (1/2)\left(\widetilde{u_i u_i} - \widetilde{u}_i \widetilde{u}_i\right)$:

$$\widetilde{D}_t \langle\rho\rangle\, k_\Delta = \partial_j(\mu + \mu_\Delta)\,\partial_j k_\Delta - \widetilde{\sigma}_{ij}\partial_j \widetilde{u}_i - \frac{C_\epsilon \langle\rho\rangle\, k_\Delta^{3/2}}{\Delta}, \tag{16}$$

and the subgrid scale viscosity $\mu_\Delta$ can then be evaluated as follows:

$$\mu_\Delta = C_k \langle\rho\rangle \Delta \sqrt{k_\Delta}, \tag{17}$$

in which both $C_k$ and $C_\epsilon$ are dimensionless single valued constants set to 0.094 and 1.05 respectively.

## 2.2 Thermophysical and Chemistry Models

In the current study, the photostationary NO - NO$_2$ - O$_3$ mechanism (Leighton, 1961) will be considered. The mechanism comprises the photodissociation of NO$_2$ into NO, the titration of O$_3$ by NO to form NO$_2$, and an intermediate third-body reaction for in situ ozone production through oxygen radicals originated from NO$_2$ photolysis:

$$\text{NO}_2 + h\nu \rightarrow \text{NO} + \text{O}(^3\text{P}) ; \quad \lambda \leq 420\,\text{nm} \tag{R1}$$

$$\text{O}(^3\text{P}) + \text{O}_2 + \text{M} \rightarrow \text{O}_3 + \text{M} \tag{R2}$$

$$\text{NO} + \text{O}_3 \rightarrow \text{NO}_2 + \text{O}_2 \,. \tag{R3}$$

Reaction (R1) describes the photodisocciation of NO$_2$, whose rate constant $k$ is calculated as a function of the solar zenith angle ($\chi$):

$$j_{R1} = \begin{cases} J_0 \cos^m(\chi) \exp\left[\dfrac{-n}{\cos(\chi)}\right] , & |\chi| < 90° \\[2mm] 0 , & \text{otherwise.} \end{cases} \tag{18}$$

Meanwhile, the rate constants for reactions (R2) and (R3) are based on the Arrhenius rate law:

$$k_{\text{R}i} = A_i \exp\left[-(T_A)_{\text{R}i}/T\right], \tag{19}$$

where $A$ and $T_A$ are the corresponding pre-exponential factor and activation temperature for each reaction. All parameters for the aforementioned reactions are tabulated with reference in Table 1.





In the current study, air is modelled as a mixture of its primary constituents, namely, $N_2$ and $O_2$, but the effects of $H_2O$ and $CO_2$ are not considered at this stage. Together with the addition gaseous species introduced in reactions (R1) to (R3), that is

NO, $NO_2$, $O(^3P)$, and $O_3$, Equation 14 consists of six species transport equations that need be solved. The individual species net production rate $r_q$ in equation (14) is determined by summing up the generation and consumption rates of the corresponding species across all relevant reactions. As non-limiting examples, the net production rates for NO, $NO_2$, and $O_3$ for the current chemical mechanism are calculated as follows:

$$r_{NO} = j_{R1} \left[ NO_2 \right] - k_{R3} \left[ NO \right] \left[ O_3 \right] \tag{20}$$

$$r_{NO_2} = -j_{R1} \left[ NO_2 \right] + k_{R3} \left[ NO \right] \left[ O_3 \right] \tag{21}$$

$$r_{O_3} = k_{R2} \left[ O(^3P) \right] \left[ O_2 \right] \left[ M \right] - k_{R3} \left[ NO \right] \left[ O_3 \right]. \tag{22}$$

Transport and thermophysical properties for these species are available from McBride et al. (1993). Temperature dependence of specific heat for species $q$ at constant pressure ($c_p^q$) is presented as a polynomial function:

$$c_p^q = R_q \sum_{i=0}^{4} a_i T^i, \tag{23}$$

where the polynomial coefficients $a_i$ are obtained using least-square fit, and $R_q$ is the specific gas constant for the species of interest. Meanwhile, dynamic viscosity ($\mu$) is represented using Sutherland's law:

$$\mu = \frac{A_S T^{1/2}}{1 + \dfrac{T_S}{T}}, \tag{24}$$

where $A_S$ (in Pa·s·K$^{-1}$) and $T_S$ (in K) are species-specific single-valued coefficients. The thermodynamic state is determined by the ideal gas relation:

$$p = \rho R T, \tag{25}$$

where $R$ is the specific gas constant of the local gas mixture.

## 2.3   Model Implementation

The open source finite volume computational continuum mechanics framework OpenFOAM (Weller et al., 1998) has been used as the modelling foundation for this work, due to its ability to handle compressible reactive flows in an unstructured grid ar-

rangement. More specifically, the current flow solver and related data processing utilities are derived from OpenFOAM version 7, maintained by CFD Direct Ltd.. All pertinent modifications and augmentations of at the source level are performed in the C++ programming language. The OpenFOAM framework contains core classes for LES and gas-phase thermophysics, chemistry and reaction kinetics. To adopt OpenFOAM for urban-scale chemical transport modelling, a custom solver, `urbanChemFoam`, derived from the existing `rhoReactingFoam`, has been developed to enable geodetic and solar calculations using a Julian

Day based date/time module. The solar state, represented by the solar zenith angle ($\chi$) can either be calculated at each time



step, using the algorithm of Meeus (1998), via a user-defined reference time and location, or set to a fixed value by the user throughout the model run.

Additional components have also been developed to operate in conjunction with `urbanChemFoam`. As existing chemical reactions in OpenFOAM are functions only of the thermodynamic state, an new reaction class, called
`irreversiblephotolysisReaction`, has been developed to model photodissociation as a function of the cosine of the solar zenith angle. Further, surface emissions for instance from traffic sources are provided in units of mass flux (i.e., kg·s$^{-1}$). These are typically introduced in chemical transport models as volumetric source terms in the interior cell nodes of the corresponding surface boundaries. On the other hand, in OpenFOAM, emissions must rendered as a flux of mixing ratios (in kg·s$^{-1}$ per kg mixture). Since the mass of the mixture, and therefore the corresponding mixing ratio flux, is not known
at the boundary surface, this necessitates the implementation of a new boundary condition, `simpleEmission`, in which the mixture mass at the boundary is calculated using the corresponding interior boundary cells, allowing surface emissions to be prescribed as mass flux boundaries. To prevent spurious accumulation of pollutant species through the spanwise cyclic boundaries, the default `cyclic` boundary patches in OpenFOAM have been augmented to behave as a Neumann boundary condition (`zeroGradient`) on the downstream boundary of the flow, while maintaining background concentrations on the
upstream boundary, similar to a Direchlet boundary condition (`fixedValue`). These modified boundary patches are referred to as `cyclicZeroGradient` and `cyclicFixedValue` respectively. Finally, a preprocessing utility, `initCanyon`, has been developed to provide an initial velocity perturbation in the solution domain, allowing turbulence flow structures to freely develop between the spanwise and lengthwise cyclic boundaries.

### 2.4 Domain Discretization and Decomposition

The solution domain is disretized using the OpenFOAM `blockMesh` utility, which generates a conformal hexahedral grid. A *core* region is defined by a 9 m lengthwise section about the origin, covering the entire span and height of the street canyon, where grid cells are uniform and have near unity aspect ratio. Grid cell grading is introduced beyond the core region, so that cell size gradually expand at a rate according to a user-specified final expansion ratio, relative to the core cell size, dictating the size of the furthermost cells. For the present study, this expansion ratio is set to 1:2 in the spanwise direction, 1:3.2 lengthwise
and 1:4 in height. Two mesh configurations of core cell sizes of 1 m (*coarse* mesh) and 0.5 m (*fine* mesh) have been deployed in the evaluation study, corresponding to a total cell count of 43 500 and 344 000 cells. Following the results from the stationary evaluation (Section 3), the transient study is conducted only at the fine mesh level. Figure 2 shows the mesh cell distribution for the fine mesh arrangement (core cell size 0.5 m), showing core and graded regions.

The numerical schemes used in the discretization of the prognostic equations are second-order accurate in time and space. A
collocated grid approach is used, in which pressure-momentum coupling is achieved using a fourth-order interpolation scheme (Rhie and Chow, 1983). For each time step, each discretized prognostic equation is solved interatively using the biconjugate gradient method (van der Vorst, 1992) with a diagonal incomplete LU-conditioner, until the global normalized residual falls below a specified convergence criterion, in this case $10^{-5}$. Mass conservation is enforced at every iteration by solving an iterative pressure correction equation until the aforementioned residual threshold is reached. Typically, the pressure correction





equation converges in two to three iterations, and about five to eight iterations for the prognostic equations at each time step for the sizes of grid cells and time step considered in this study. Grid decomposition for parallel computing is performed in three dimensions using a dual recursive bipartitioning algorithm (Chevalier and Pellegrini, 2008).

## 3  Stationary Evaluation

For the purpose of evaluating the LES street canyon model, a stationary run is performed for which LDA flow measurement
data are available from Li et al. (2008) for a scale model infinite canyon arrangement, also having an height-to-width aspect ratio of two, conducted in a water tunnel. The horizontal velocity of this simulation is prescribed at the freestream boundary, denoted as $U_\infty$. An initially quiescent velocity field is imposed in the canyon region, while a linear spanwise flow profile – that is, $u(z) = U_\infty (z - H) / (z_\infty - H)$, where $z$ is the elevation above the canyon, $z_\infty$ and $U_\infty$ is the elevation and velocity at the domain freestream boundary respectively – is assigned above the canyon. A value of $3.52\,\text{m·s}^{-1}$ has been used for the spanwise
velocity at the freestream boundary. Turbulence in the flow domain is initiated using a superposition of randomized velocity perturbation, with magnitude up to 10% of the freestream velocity, into the initial flow field. Simulations are conducted using with the coarse (1 m) and fine (0.5 m) mesh configurations, both at a uniform time step of 0.01 s. Each simulation is given a six-hour (21 600 s) spin-up period before sampling takes place, where the flow field is sampled for 3 600 s at 100 Hz for constructing turbulent statistics.

To coincide with the LDA measurements, sampling points for the LES simulations are located along vertical stations at $x/W$ = -0.25, 0, and +0.25, denoting upstream, central, and downstream positions with respect to the direction of freestream flow, at $y = 0$, the lateral plane of symmetry. The time-averaged velocity components and subgrid scale turbulent kinetic energy (SGS TKE) at each sampling point are calculated as the arithmetic mean of the respective time series. The turbulent velocity fluctuation in each direction is, in turn, determined using the L2-norm of the central difference of the instantaneous velocity
components. The resolved scale TKE can then be calculated using its definition $k = (1/2)\,(\widetilde{u}_i \widetilde{u}_i)$. Note that the LDA mean and fluctuation measurements represent a Reynolds average of the flow field, while the LES results in the present study attempt to duplicate these results using only data contained in the resolved scale.

Figure 3 shows the comparison between the present LES results on both mesh arrangements with the LDA measurements at the aforementioned vertical stations. Both the fine and coarse mesh results are in excellent agreement with respect to horizontal
time-averaged velocity ($\overline{u}$, top left panel), with the exception of a slight overprediction above the canyon at $x/W = \pm\,0.25$. For the time-averaged vertical velocity component ($\overline{w}$, upper right panel), the fine mesh solution is able to adequately capture the profiles. There is a discrepancy at $x/W = 0.25$, but the salient features are still well represented. In comparison, the coarse mesh results underpredict the velocity magnitude. In light of the fine mesh results, this observation indicates an issue with grid convergence. In addition, substantially better agreement in turbulent fluctuation ($u'$ and $w'$, lower left and right panels,
respectively) can be found with the fine mesh results. There are still minor differences between the fine grid LES results and the LDA measurements, but can be attributed to the use of only resolved turbulent fluctuations in deriving the time-averaged results. In addition, as Zhong et al. (2015) also indicated, the circumstances under which the experiment (Li et al., 2008) is





conducted, methods of vortex generation for instance, are not modelled, which could attribute, at least in part, to the differences observed.

The mean flow field in the canyon along the lateral plane of symmetry has been obtained by averaging the instantaneous velocity field at interval of 300 s for over the 3600 s sampling period for the fine mesh configuration. It is presented in Figure 4 using a line integral convolution representation (Cabral and Leedom, 1993). It depicts a typical skimming flow regime occurring in idealized urban canyon arrays (Oke, 1988), where a free shear layer develops over the canyon at $z/H = 1$ resulting from turbulent interaction between the building roof and the skimming flow upstream. Two vertically stacked primary recirculation

zones exist inside the canyon, in line with existing works on deep street canyons (Baik and Kim, 1998; Li et al., 2008). However, additional, minor vortices are also observed at the corners at the street level, which may lead to additional isolation of street level emissions, for instance, from traffic sources.

      Although, in the skimming flow regime, there is little mass transfer between the bulk flow and the canyon, as indicated by the by the RANS vertical velocity component ($\overline{w}$) near the canyon roof in Figure 3. Based on the mean flow structures indicated

in Figure 4, weak vertical interaction still exists downstream in the canyon between the primary recirculation zones. It can be seen that pollutants from the street level can be carried upwards by the bulk flow, first through the lower recirculation zone in an anticlockwise direction downstream. Further exchange then takes place between the lower and upper recirculation zones, eventually reaching the canyon roof clockwise around the upper recirculation zone, where it merges with the bulk flow, again downstream in the canyon.

To further evaluate the resolution of turbulent scales under each mesh configuration, the fraction of TKE as its simulated total – that is, the sum of resolved and SGS TKE – is calculated along the three vertical stations, as presented in Figure 5. The free shear layer on top of the canyon roof region is accompanied by a sudden increase in turbulence kinetic energy upstream in the canyon core, which dissipates downstream (Driver and Seegmiller, 1985). However, a decrease in resolved scale TKE fraction can be observed in this region. In particular, the attenuation in turbulence resolution is significantly higher in the

coarse mesh than in the fine mesh configuration, with a minimum of 36.0% and 73.6% for the respective mesh density. This indicates a diminished capability of the coarse mesh to adequately capture turbulent structures explicitly, which contributes to the discrepancies in the vertical profiles observed in Figure 3. Nevertheless, the resolved TKE fraction in the canyon core and the bulk flow regions (that is, outside of the free shear flow region) remains very high for both grid configurations. Based on a 95% confidence interval for regions away from walls and the free shear layer, the resolved fraction is calculated to be

86.0% - 87.5% for the coarse mesh, and 92.6% - 93.3% for the fine mesh. This suggests that the turbulent resolution in the free shear region could be summarily resolved by local mesh refinement. However, based on Figure 5, the fine mesh configuration provides an adequate resolution of turbulent flow structures. Hence subsequent discussions will be made exclusively with the 0.5 m mesh configuration.





## 4 Transient Study with Photostationary NO - NO$_2$ - O$_3$ Mechanism

The transient study consists of two sets of model runs. The first model set is a fully transient run, where boundary conditions are prescribed using the diurnal profiles for meteorological conditions, solar state, background concentrations, and traffic emissions. For the second set, series of stationary runs are conducted with boundary conditions fixed at 8:00, 12:00, 16:00, and 20:00 UTC. Each run begins six hour prior to the start of the data sampling period, and are conducted using the fine mesh arrangement (core cell size 0.5 m) and a constant time step of 0.05 s. Vertical profile data are extracted at $x/W$ = -0.25, 0, and

+0.25, as well as in-canyon horizontal profile data at $z$ = 2 m, $z/H$ = 0.5 and 075.

### 4.1 Model Configuration

Figure 6 shows the diurnal profiles for meteorology, background concentrations, traffic emissions, and solar state applied to the transient model run, representative of a typical work day for the City of Berlin in July 2014. The freestream horizontal ($u$) wind, as well as the solar zenith angle ($\chi$) is presented in Figure 6(A), and the air temperature and pressure are illustrated in

Figure 6(B). The profiles for wind, pressure and temperature have been obtained from the hourly WRF model output of Kuik et al. (2018) by first interpolating the pressure coordinates to the freestream height (112 m) of the canyon and averaging them at each hour across every day in the period of July 2014. The solar zenith angle is calculated on-line at each time step. Sunrise and sunset are defined when the solar zenith angle reaches the 90° crossing. For the 24-hour simulation period, sunrise occurs at 02:55:06.83 UTC, and sunset takes place at 19:25:26.13. In the mean time, the solar zenith angle reaches a minimum of

29.42° at 11:10:24.67.

The profiles for background concentrations of NO, NO$_2$, and O$_3$, shown in Figure 6(C), are obtained from hourly averages on all full working days (Tuesdays and Wednesdays; Mondays and Thursdays have been excluded to minimize end-effects carried over from adjacent non-working days) from observation data of urban background stations located on Amrumer Straße (DEBE010) and Nansenstraße (DEBE034) in Berlin. Days with missing measurement data are discarded entirely, leaving 17

days with intact hourly measurements. The traffic NO and NO$_2$ emissions, as presented in Figure 6(D), are estimated from annual NO$_X$ aggregate for the city of Berlin in 2014 using hourly traffic data and vehicle fleet composition provided by the Berlin City Senate. Temporal redistribution of annual traffic emissions are inferred by traffic density for major street segments, from which a representative flow speed and level of service are determined. Emission factors for NO$_X$ can then be calculated assuming a standard fleet composition for the street segment of interest. The NO$_X$ emission is them partitiioned into NO and

NO$_2$ according to the Gridding Nomenclature for Reporting (GNFR) for the road transport source sector categories (Cranier et al., 2019), with molar fraction of 0.95 for NO and 0.05 for NO$_2$ for gasoline exhaust (F1), and 0.7 (NO) and 0.3 (NO$_2$) for diesel exhaust (F2).

All runs are conducted with four cluster nodes, each comprising a dual Intel Xeon Platinum 9242 processors and 384 Gb of physical memory. Based on OpenFOAM run statistics, each simulation hour of model run requires on average 8623 ± 148 s in

processor time, or 8788 ± 145 s in wall clock time to complete, with a 95% confidence interval over all transient and stationary





runs. All output data are written to file every 30 seconds of simulation time, and corresponding hourly means are calculated at the top of each hour using a central window average spanning 3600 s.

### 4.2 Results and Discussion

Figure 7 shows the qualitative comparison based on the LIC representation of the hourly average velocity field between the
fully transient run and the stationary runs at each specific times. As with Figure 4, two vertically stacked primarily recirculation regions are visible in all cases, as well as secondary vortices located at the street level corners. Differences in fluid motions can be observed in the fluid motions between the transient and stationary model runs at the same time, as indicated by the relative displacement in the primary vortex locations. This is a result of the differences in temporal variations in the meteorological conditions, which are held constant at specific times in the stationary runs, as indicated in the difference in hourly vertical
profile of horizontal velocity component in the freestream region ($z/H > 1$) between the transient and stationary runs, as shown in Figure 8. The momentum exchange in and out of the canyon is restricted by the shear region, defined by a strong mean shear ($\partial_z \bar{u}$) at the canyon roof ($z/H = 1$). The in-canyon ($z/H \leq 1$) velocity component profiles follow a general feature proportional to the magnitude of the freestream velocity. Nevertheless, while the diurnal variation in freestream velocity is approximately 2 m·s$^{-1}$, as illustrated in Figure 6(A), the corresponding change in the in-canyon velocity magnitude is in only
in a few centimeters.

However, deviations of up to about 20 ppb in pollutant concentrations can be observed in the vertical profiles in Figure 8, even though the rate of hourly traffic NO and NO$_2$ emissions are identical for the indicated times between transient and stationary simulations. The discrepancy is particularly large at 08:00 and 20:00, This is due to a two factors attributing to the difference between the two types of model runs. First, in addition to meteorological conditions, the rates of reactions,
background concentrations and emissions are also held constant in stationary simulations, while the transient run follows a diurnal variation, which results in a difference in equilibrium composition in the street canyon and freestream regions. This also brings about the second factor, that is, the difference in boundary conditions also leads to a difference in cumulative pollutant concentration at the beginning of the sampling period.

A cascade of concentrations for all species considered can be identified, corresponding to the individual primary vortices
as seen in Figure 4, concurring with the numerical results of Bright et al. (2011) for a street canyon of unity aspect ratio. The concentrations of NO and NO$_2$ generally increases along each level of cascade (and hence each primary vortex), due to the higher flow velocity in the outer regions of each vortex, causing pollutants to be transported first to the top of the vortex. A portion of NO and NO$_2$ is then transferred at the top of the lower vortex into the bottom of the upper vortex through shear motion. Though the concentration of NO and NO$_2$, the bulk motion in the upper vortex leads to an increase in vertical
concentrations along the vortex, though in lower concentrations, before leaving the canyon through the shear layer at the roof level.

In addition, there is a reciprocal relationship between the concentrations of NO and NO$_2$, and O$_3$. This is expected as the O$_3$ originates from the freestream and is titrated through reaction R3 by the NO originating from street level traffic. However, the concentration of in-canyon O$_3$ is not substantially different from freestream levels, at least in comparision with the NO and





NO$_2$ levels. This is, in part, due to the in situ production O$_3$ through photodissocation in reaction R1. The exception is 20:00, where the hourly sampling and averaging takes place with the absence of the photolysis reaction R1, as the solar zenith angle ($\chi$) reaches 90° at 19:25, leaving the titration reaction R3 the only active reaction in the mechanism. Combined with a low traffic emission, as indicated in Figure 6(D), this leads to a moderate decrease in O$_3$ while depleting the NO in the canyon.

Horizontal profiles of velocity and concentrations are presented in Figures 9, 10, and 11, evaluated at 2 m, 18 m ($z/H = 0.5$), and 27 m ($z/H = 0.75$) above the canyon street level respectively, representing street level, between the two vertical primary vortices, and the upper primary vortex. The direction of each vortex is indicated by the mean horizontal and vertical velocity components, with the lower vortex travelling anticlockwise, and clockwise for the upper vortex. There are slight differences (within $10^{-2}$ m) in velocity distributions between the transient and corresponding stationary runs, although both sets of runs follow a similar velocity profile. The velocity magnitude also increases with elevation as it approaches freestream flow.

The stratification of NO$_X$ at each horizontal level follow the direction of the vortex at the respective elevation, with higher concentrations level found towards the right wall in the lower vortex (Figures 9 and 10), and the left wall in the upper vortex (Figure 11). However, at the 2 meter level, the peaks of NO$_X$ concentration resulting from diffusion of street level emissions, as depicted in Figure 1(C), is not evidently shown. This suggests dominance of spanwise advection near the street surface in the lower vortex, which results int the aforementioned high concentration region towards the right side of the lower vortex. By comparison, the concentration of O$_3$ remains more stable at all horizontal levels.

Finally, Figure 12 shows the time series in the concentrations of NO, NO$_2$, and O$_3$ in the street canyon from the transient model run, at nine different locations corresponding the intersections between the vertical (Figure 8) and horizontal profiles (Figures 9 to 11). Both the NO and O$_3$ profiles exhibit a typical dirunal cycle, which show strong dependence with solar activity As previously pointed out, as before sunrise and after sunset (that is, $|\chi| \geq 90°$) only the titration reaction R3 remains active. Thus, in conjunction of emission rates from traffic NO$_X$ during these times, a drastic reduction in NO can be seen. Further, the O$_3$ concentration at all monitoring points follow the same tendency as the background O$_3$ shown in Figure 6(C), except at the time immediately following the cessation of solar activity (at 19:25), where an acute reduction is observed due to NO$_X$ still presently available, as well as from emission sources, for the titration.

While the trend of NO$_2$ concentration in the afternoon and early evening (that is, between 15:00 and 21:00) is in line with the traffic emissions, the overall diurnal variation does not reflect the morning traffic peak, but instead only gradually increases following sunrise at 02:35. As the morning period is accompanied with a comparatively low background O$_3$ concentration, the photodissociation of NO$_2$ becomes the dominant reaction in the mechanism, leading to consumption of the emitted NO$_2$ from the traffic sources. As the O$_3$ concentration increases in the afternoon, the rate of titration increases. Therefore the afternoon NO$_2$ peak coincides with a decrease in NO concentration.

# 5 Concluding Remarks

A weakly compressible, reactive finite volume LES solver, based on the OpenFOAM computational continuum mechanics framework (Weller et al., 1998), has been developed for modelling chemical transport of gas-phase pollutant species in an





idealized street canyon under conditions where meteorology, background pollutant concentrations, and traffic emissions vary with time. A deep street canyon with a height-to-width aspect ratio of two (Zhong et al., 2015), has been used throughout for

this study. The solver has been evaluated using stationary LDA measurements from Li et al. (2008), showing excellent agreement in mean and turbulent fluctuations. Whereas previous studies on street canyons have been conducted under stationary conditions, a subsequent transient model run with chemical transport, time-varying boundary conditions representing a typical working day for the City of Berlin in July 2014 covering a 24-hour period have been prepared. Comparisons have also been made with stationary runs corresponding certain points of time during the diurnal cycle. While vertical and horizontal profiles

for velocity components indicate minor differences between the stationary runs and the transient runs at corresponding times, a large deviation in pollutant species concentration can be observed. This discrepancy is caused by time variations in ambient conditions, which also leads to differences in cumulative concentrations prior to solution sampling, both factors not considered in stationary runs. Diurnal variations in pollutant concentrations can also be inferred through the transient model run at select locations in the canyon, showing typical profiles of NO, $NO_2$, and $O_3$ that correspond to traffic emissions, background

concentrations, as well as solar zenith angle.

The results of this study demonstrate the ability for this solver in performing urban-scale chemistry transport modelling at both a stationary and non-stationary capacity in a grid-independent manner. As such, it can be readily adapted to model a larger domain with more realistic geometry, such as a city block. For models covering such domain regions, however, input data such as meteorology and emission profiles need be prescribed at higher spatial and temporal resolutions, which can be accomplished

coupling the model with outputs results from regional models, WRF for example. Further, the method of temporal-spatial distribution and $NO_X$ emissions from annual aggregate, and the subsequent partition into NO and $NO_2$ contributions, as outlined in Section 4.1, can be extended to cover a road network, where emissions can be mapped as discretized surface fluxes in a similar manner as the `simpleEmission` boundary condition described in Section 2.3.

In addition to $NO_X$, the emissions of PM, carbon monoxide (CO), volatile organic compounds (VOCs) can be processed

in a similar way. While the photostationary NO - $NO_2$ - $O_3$ mechanism (Leighton, 1961) serves as the integral component of the OpenFOAM chemistry model for the current study, modelling transformation processes of CO and VOCs will certainly require chemical kinetics that include reactions of these species and their intermediate derivatives, such as OH, R-H, and R-$O_2$. Mechanisms such as CBM IV (Gery et al., 1989), involving 32 species and 81 reactions, or, to a lesser degree, the SMOG mechanism (Damian et al., 2002), made up of 13 species and 12 reactions, can be considered in subsequent investigations.

**Nomenclature**

*Roman Symbols*

| | |
|---|---|
| $a_i$ | Polynomial coefficient for the $i$th power term |
| $A$ | Arrhenius pre-exponential factor, in units of [m, kmol, s, K] |
| $A_S$ | Sutherland law reference viscosity factor [kg·m$^{-1}$·s$^{-1}$·K$^{-1/2}$] |
| 430 $C_k$ | Parametric constant for subgrid scale viscosity [] |




| $C_\epsilon$ | Parametric constant for SGS TKE dissipation rate [] |
| $c_p$ | Specific heat at constant pressure [J·K$^{-1}$·kg$^{-1}$] |
| $D$ | Binary diffusion coefficient [kg·m$^{-1}$·s$^{-1}$] |
| $Da$ | Damköhler number [] |
| $e$ | Internal energy [J] |
| $h$ | Enthalpy [J] |
| $h$ | Planck constant (in reaction R1) [J·s] |
| $i, j, k, q$ | Generic indices |
| $J_0$ | Photolysis rate at $\chi = 0$ [s$^{-1}$] |
| $j_R$ | Photolysis rate [s$^{-1}$] |
| $k$ | Turbulent kinetic energy [m$^2$·s$^{-2}$] |
| $k_R$ | Reaction rate constant [m$^3$·kmol$^{-1}$·s$^{-1}$] |
| $m, n$ | Photolysis rate parameters [] |
| $p$ | Pressure [Pa] |
| $R$ | Specific gas constant [J·K$^{-1}$·kg$^{-1}$] |
| $r$ | Net species production rate [m$^3$·s$^{-1}$] |
| $S_{ij}$ | Shear strain rate tensor [s$^{-1}$] |
| $T$ | Temperature [K] |
| $T_A$ | Arrhenius activation temperature [K] |
| $T_S$ | Sutherland law reference temperature [K] |
| $U$ | Reference velocity [m·s$^{-1}$] |
| $u, v, w$ | Velocity components [m·s$^{-1}$] |
| $u', v', w'$ | Velocity fluctuation [m·s$^{-1}$] |
| $x, y, z$ | Cartesian coordinate [m] |
| $y_q$ | Mass mixing ratio for species $q$ [] |

*Greek Symbols*

| $\Delta$ | Subgrid filter length scale [m] |
| $\delta x$ | Local grid size [m] |
| $\delta_{ij}$ | Kronecker delta [] |
| $\epsilon_{ijk}$ | Permutation tensor [] |
| $\lambda$ | (Photon) wavelength [m] |
| $\mu$ | Dynamic viscosity [Pa·s] |
| $\nu$ | (Photon) vibrational frequency [s$^{-1}$] |
| $\xi$ | Cartesian coordinate (alternative to $x$) [m] |

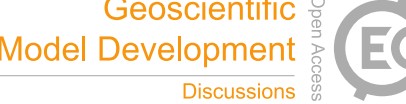

| 465 | $\rho$ | Density [kg·m$^{-3}$] |
| | $\sigma_{ij}^{\varphi}$ | Subgrid scale term for scalar $\varphi$ |
| | $\tau_{ij}$ | Shear stress tensor [Pa] |
| | $\phi$ | Latitude [] |
| | $\varphi$ | Generic scalar variable |
| 470 | $\chi$ | Solar zenith angle [] |
| | $\omega$ | Angular velocity [s$^{-1}$] |

*Mathematical Operators*

| | $d_x(\cdot)$ | Direct derivative: $d(\cdot)/dx$ |
| | $\partial_x(\cdot)$ | Partial derivative: $\partial(\cdot)/\partial x$ |
| 475 | $\partial_i(\cdot)$ | Partial derivative (shorthand): $\partial(\cdot)/\partial x_i$ |
| | $D_t(\cdot)$ | Substantial derivative: $\partial_t(\cdot) + \partial_j[u_i(\cdot)]$ |
| | $F(\underrightarrow{x}|\Delta)$ | Spatial low-pass filter kernel with cut-off length $\Delta$ |
| | $[\varphi]$ | Volumetric concentration |
| | $\overline{\varphi}$ | Temporal (RANS) average |
| 480 | $\langle\varphi\rangle$ | Spatial average |
| | $\widetilde{\varphi}$ | Density-weighted (Favre) average: $\langle\rho\varphi\rangle/\langle\rho\rangle$ |

*Acronyms and Abbreviations*

| | BLUME | Berlin Air Qualify Measurement Network (*Berliner Luftgüte-Messnetz*) |
| | CBM | Carbon Bond Mechanism |
| 485 | DNS | Direct Numerical Simulation |
| | GNFR | Gridding Nomenclature for Reporting |
| | LDA | Laser Doppler Anemometry |
| | LES | Large-Eddy Simulation |
| | PM | Particulate matters |
| 490 | RANS | Reynolds-Average Navier-Stokes (Equations) |
| | RCS | Reduced Chemical Scheme |
| | SGS | Subgrid Scale |
| | TKE | Turbulent Kinetic Energy |
| | VOC | Volatile Organic Compound |
| 495 | WRF | Weather Research and Forecasting (Model) |



*Code and data availability.* The exact version of `urbanChemFoam` and all components described in Section 2.3, as well as select test cases featured in this article, are licensed under the terms of GNU General Public License version 3.0 or later, and can be obtained from `https://github.com/echIASS/urbanChemFoam-1.0/releases/tag/gmd-2020-432`.

*Author contributions.* The activities outlined in this article have been conceived and designed by ECC and TMB. ECC is responsible for
500  the development of `urbanChemFoam` and all related components, execution of all model runs, and preparation of the manuscript. TMB provides technical and scientific inputs at all stages of the study.

*Competing interests.* The authors declare that there are no conflict of interest.

*Acknowledgements.* The BLUME urban background concentration station measurements and the aggregated annual traffic emissions data have been provided by the Senate Administration for Environment, Traffic, and Climate Protection (*Senatsverwaltung für Umwelt, Verkehr*
505  *und Klimaschutz*) of the City of Berlin. The stationary LDA measurement data of Li et al. (2008) have been made available by Drs. J. Zhong and X. Cai from the University of Birmingham. The authors would also like to express their gratitude to IASS colleagues Drs. J. Leitão, A. Lupaşcu, and J. Quedenau for their valuable inputs and support in the conception and preparation of this work.





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



**Table 1.** Reaction rate parameters for the photostationary NO-NO$_2$-O$_3$ mechanism. Units are kmol, m$^3$, s, K.

| | Rate ($k$) | Coefficients | Reference |
|---|---|---|---|
| R1 | Eq. (18) | $J_0 = 0.01165$ | Saunders et al. (2003) |
| | | $m = 0.244$ | |
| | | $n = 0.267$ | |
| R2 | Eq. (19) | $A = 2.3827 \times 10^7$ | Huie et al. (1972) |
| | | $T_A = 510$ | |
| R3 | Eq. (19) | $A = 1.8067 \times 10^9$ | Burkholder et al. (2015) |
| | | $T_A = 1500$ | |



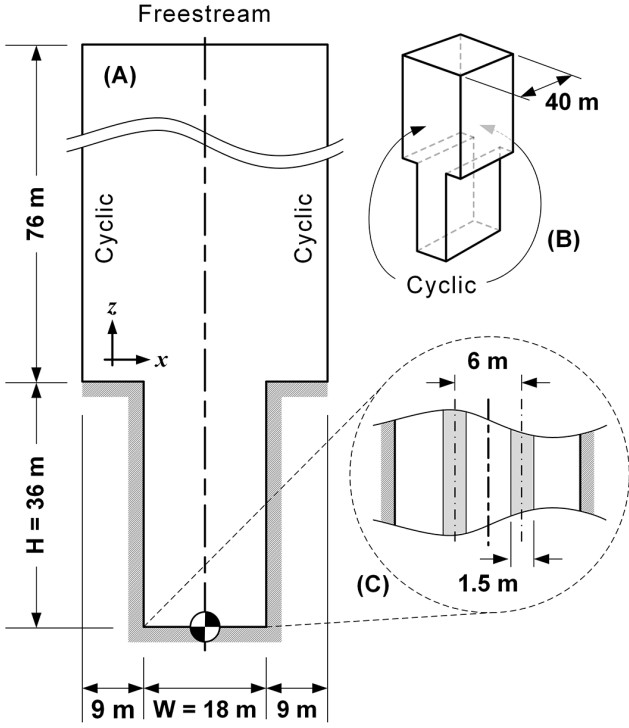

**Figure 1.** Schematic of the solution domain representing the infinite street canyon geometry from Zhong et al. (2015): (A) basic dimensions and locations of cyclic and freestream boundaries; (B) isometric view of the domain showing lateral dimension and corresponding cyclic boundaries; (C) top view of ground surface indicating size and position of simulated traffic lanes.



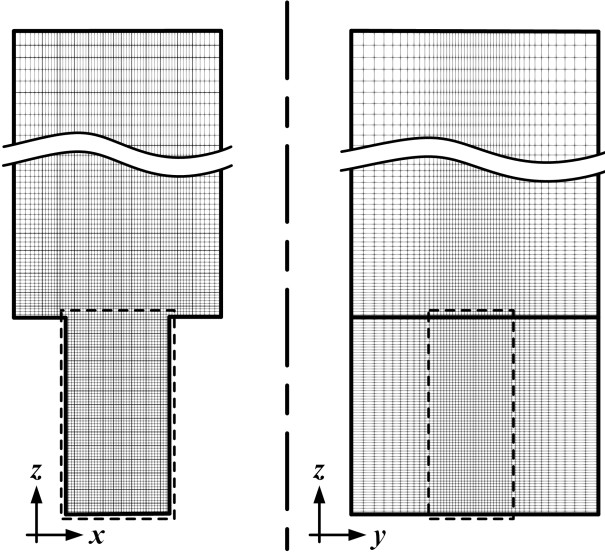

**Figure 2.** Grid distribution of the canyon geometry in (left) $xz$-plane and (right) $yz$-plane. Cell size in the canyon core region (dashed line) is $0.5 \times 0.5 \times 0.5$ m$^3$.

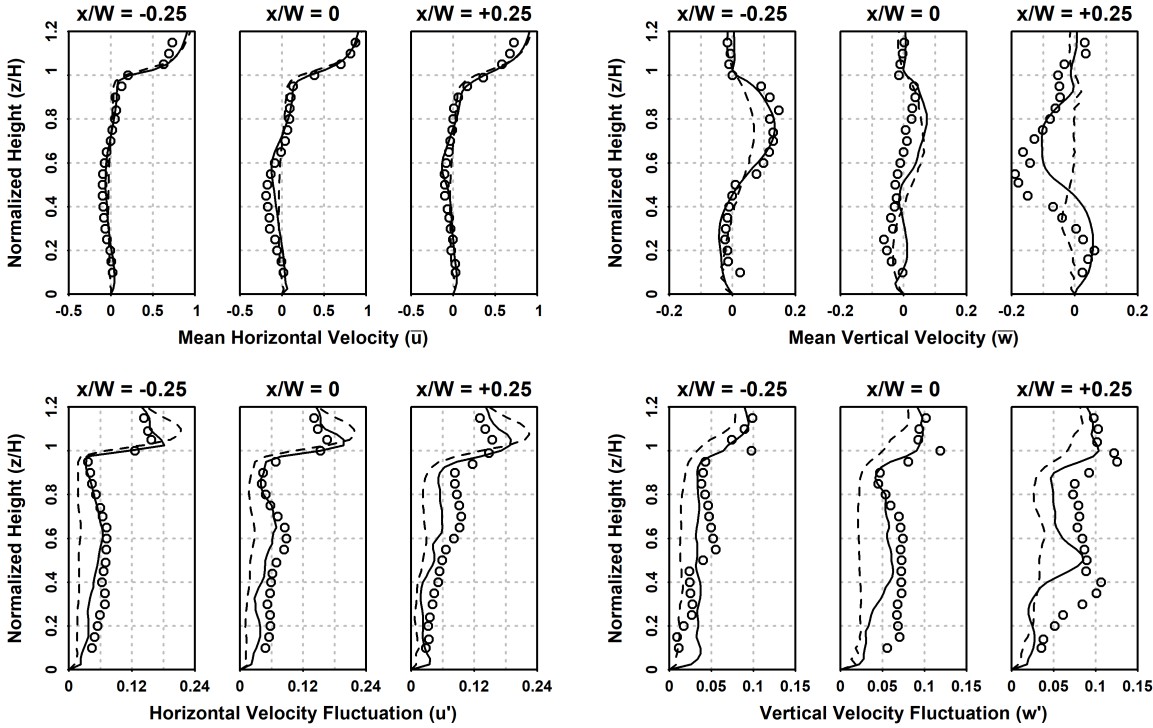

**Figure 3.** Noramlized vertical profiles of the of resolved mean horizontal (top left) and vertical (top right) velocity components, as well as horizontal (bottom left) and vertical (bottom right) velocity fluctuations, for stations located at $x/W$ = -0.5 (left), 0 (middle), and +0.5 (right) in the idealized street canyon geometry. Key: —— present study, fine mesh; - - - present study, coarse mesh; ○ LDA measurements from Li et al. (2008).





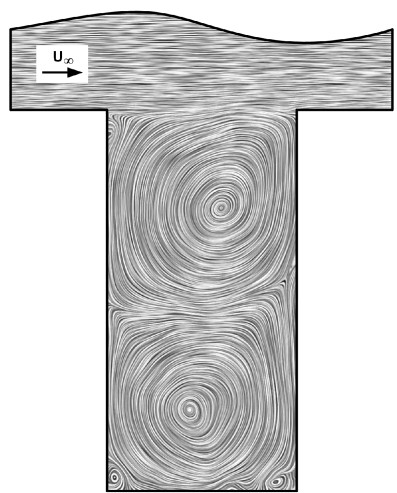

**Figure 4.** Line integral convolution representation of the velocity vector along the $xz$-plane. Inset indicates direction of freestream flow.



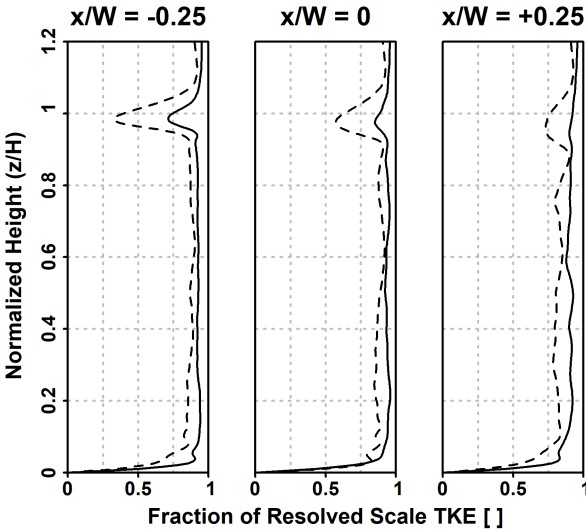

**Figure 5.** Fraction of resolved scale TKE of total (resolved and SGS) TKE for the three stations in the street canyon. Key: —— fine mesh;
- - - coarse mesh.

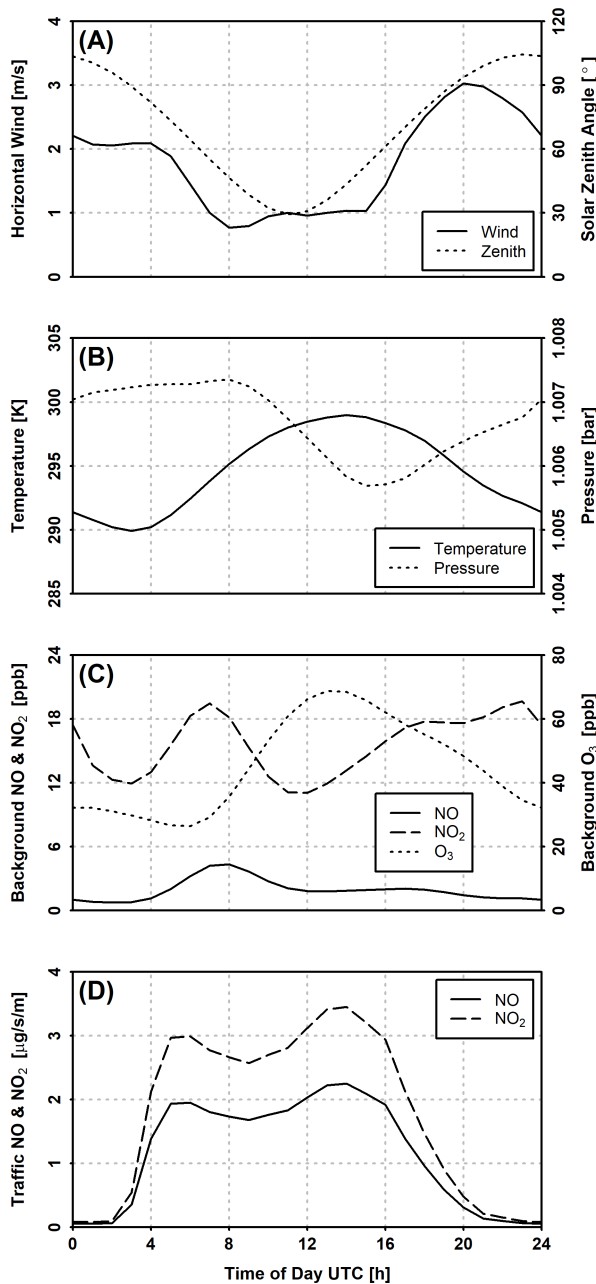

**Figure 6.** Prescribed conditions for the transient simulation: (A) horizontal freestream wind velocity and solar zenith angle; (B) ambient temperature and density; (C) urban background NO, NO$_2$, and O$_3$ concentrations; (D) workday traffic NO and NO$_2$ emissions per unit street section length.

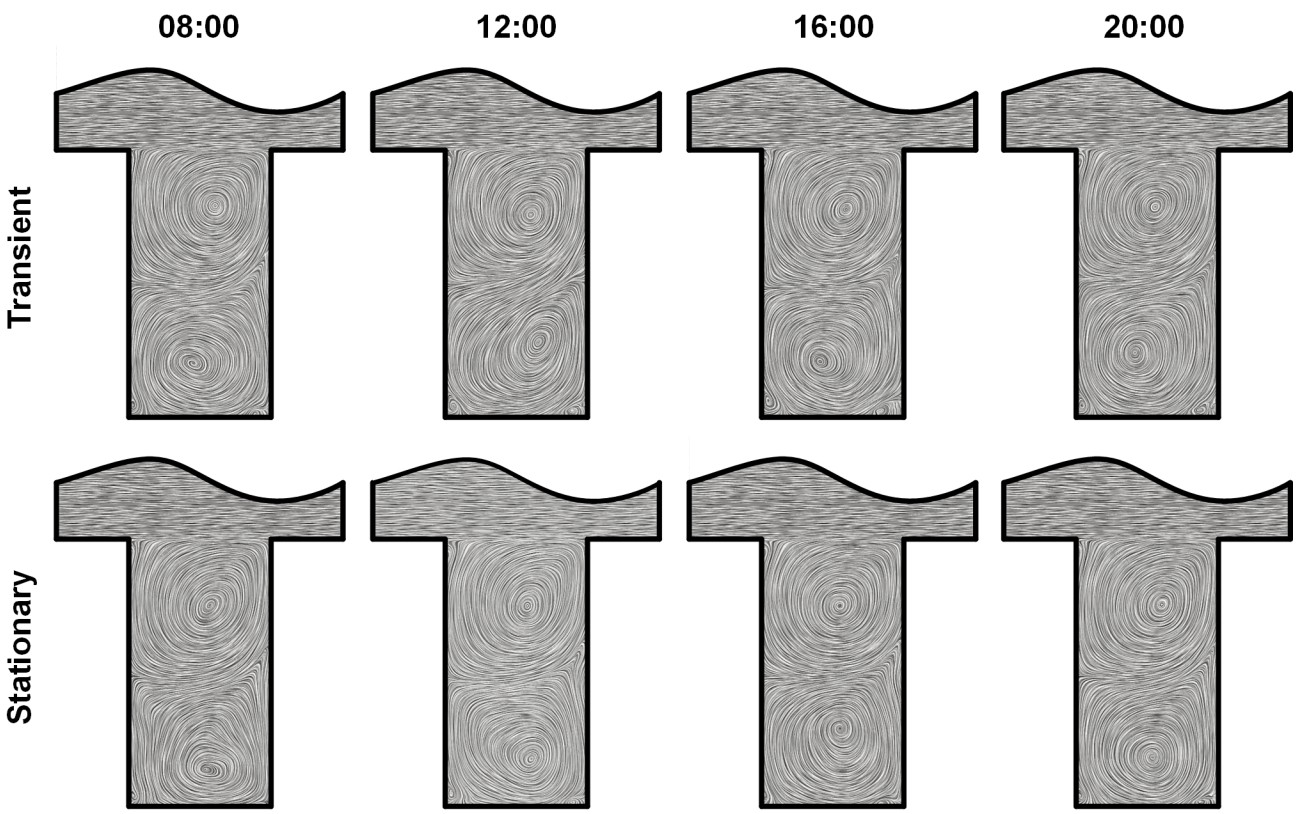

**Figure 7.** Comparison of LIC representation of hourly mean velocity vector along the $xz$-plane for (top) transient and (bottom) stationary model runs at various points of time in UTC.

**Figure 8.** Comparison of vertical station profiles of mean velocity components and species concentrations for transient and stationary model runs at various points of time in UTC. Key: —— transient; - - - stationary.



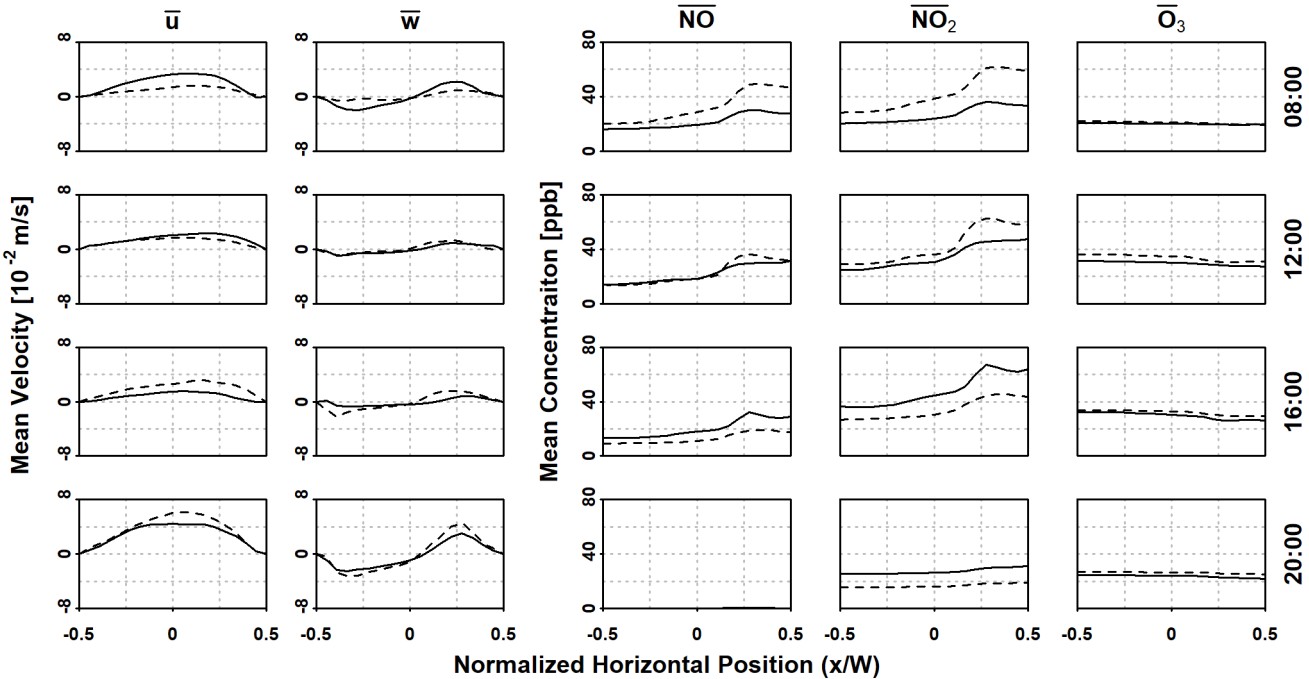

**Figure 9.** Comparison of horizontal station profiles of mean velocity components and species concentrations at an elevation of 2 m for transient and stationary model runs at various points of time in UTC. Key: —— transient; - - - stationary.





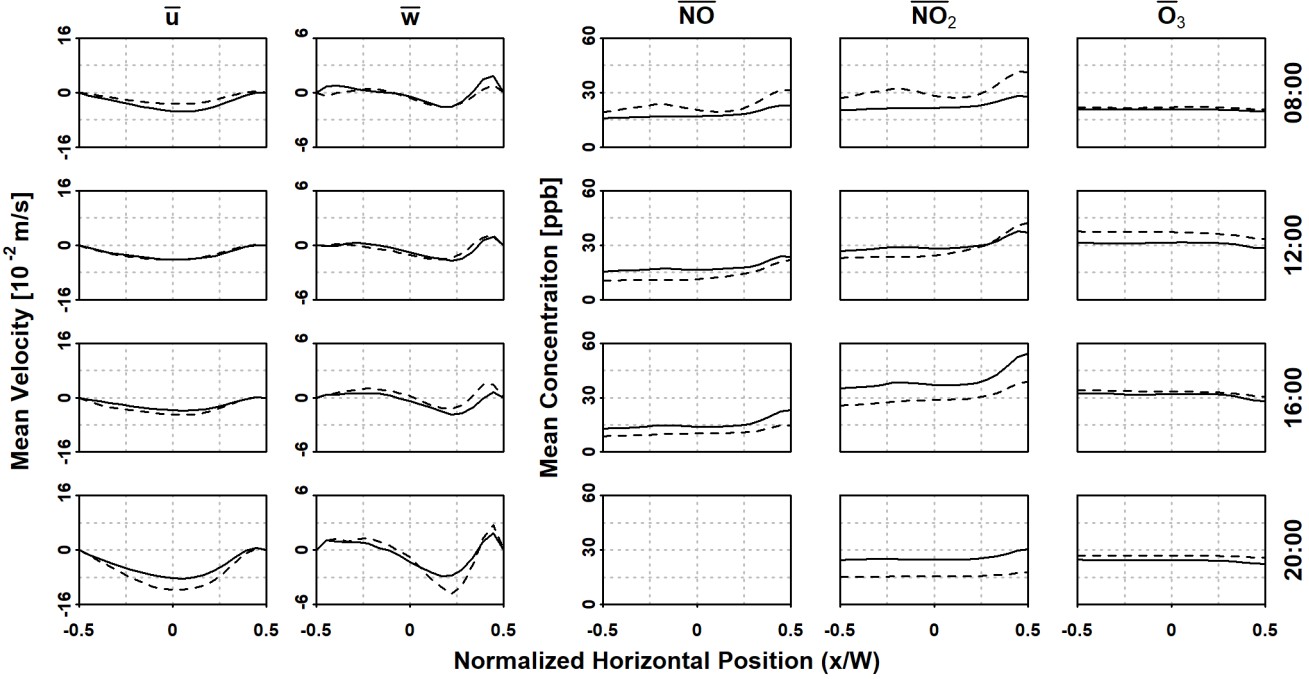

**Figure 10.** Comparison of horizontal station profiles of mean velocity components and species concentrations at a normalized elevation of z/H = 0.5 for transient and stationary model runs at various points of time in UTC. Key: —— transient; – – – stationary.



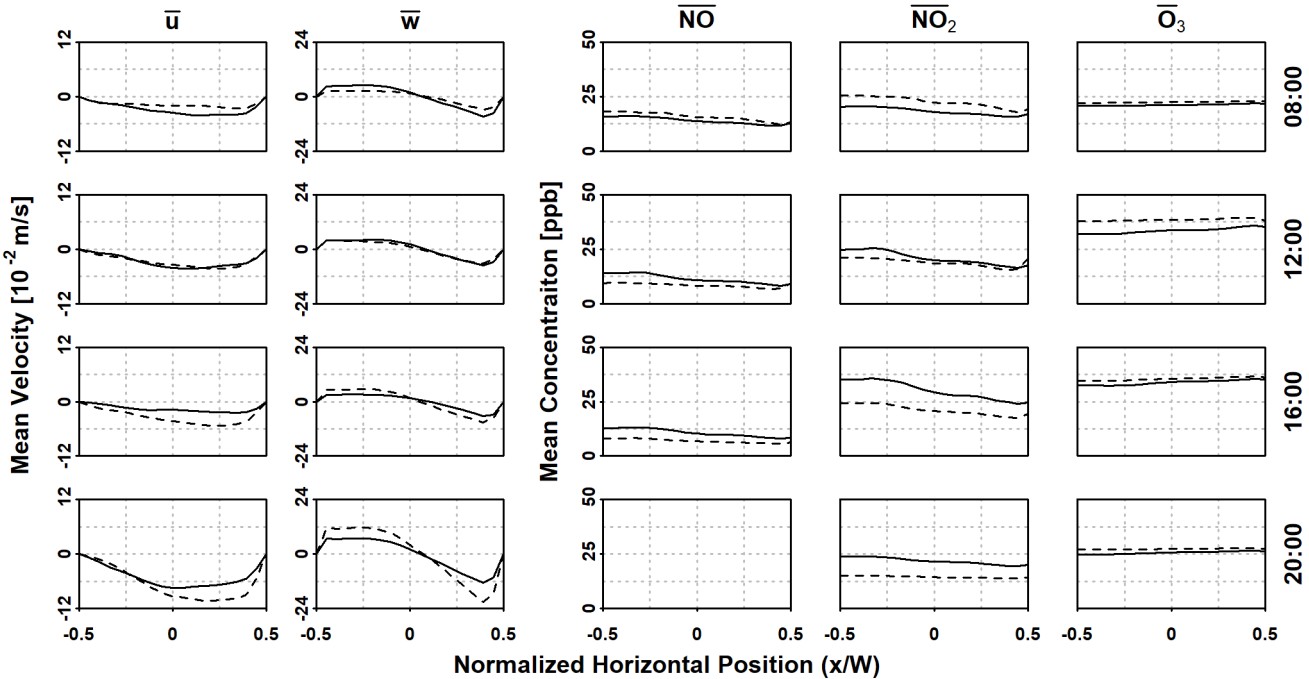

**Figure 11.** Comparison of horizontal station profiles of mean velocity components and species concentrations at a normalized elevation of $z/H = 0.75$ for transient and stationary model runs at various points of time in UTC. Key: —— transient; - - - stationary.

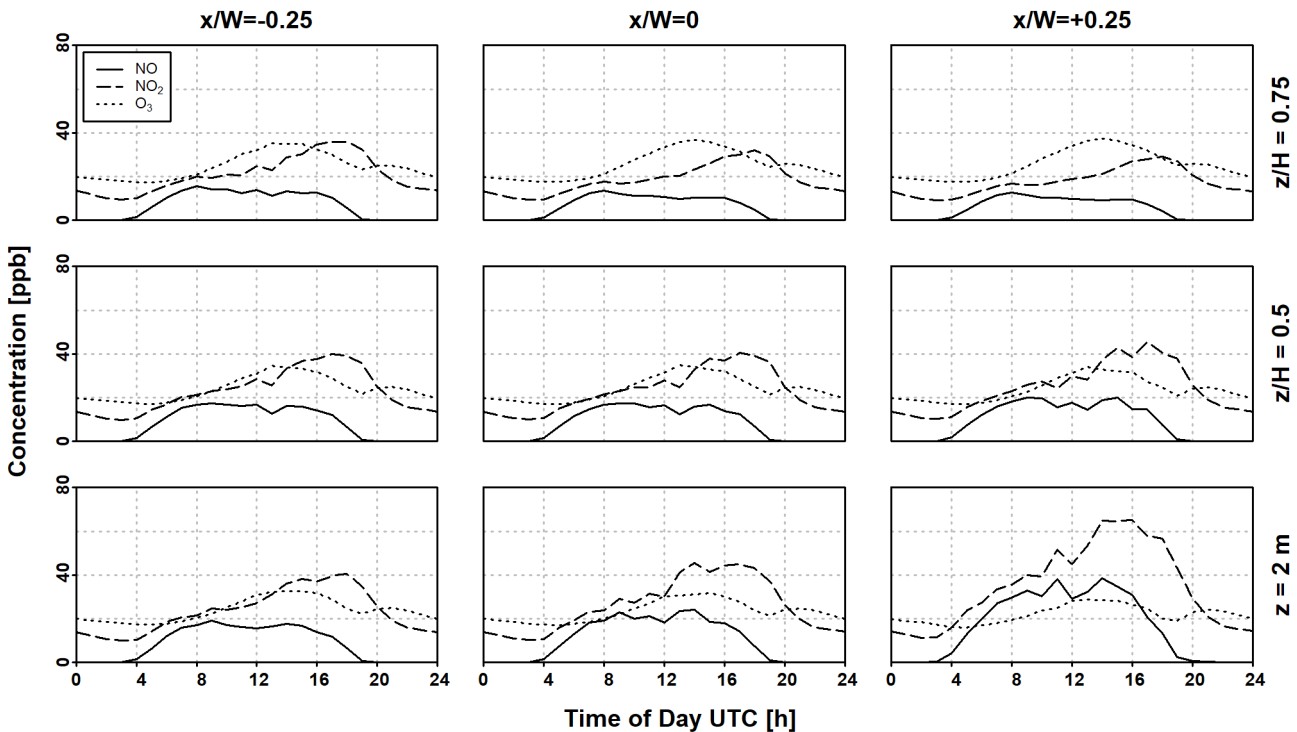

**Figure 12.** Hourly averaged time series of NO, $NO_2$, and $O_3$ concentrations at (top) z/H = 0.75, (middle) z/H = 0.5, and (bottom) z = 2m at different horizontal stations over a 24-hour period.