# Peer review of "urbanChemFoam 1.0: Large-Eddy Simulation of Non-Stationary Chemical Transport of Traffic Emissions in an Idealized Street Canyon"

_Geoscientific Model Development, 2020_

## Author Response (AR1)

**Response to Reviewer 1**

The Authors would like to thank you for the meticulous but encouraging comments on the MS. Please find our response to your comments below, with reference to your original text in italics. In the meantime, all typographical errors outlined in Minor Comments 1 - 14 have been corrected in the MS.

> **Comment:** *My main objection is that despite the interesting quantification of the discrepancies between stationary and non-stationary simulations, the study does not show the full potential of the transient setup – the transition between nocturnal and daytime boundary layer and vice versa. In particular, the morning hours would be very interesting with regard to chemistry and boundary layer processes (onset of boundary-layer growth and photochemistry, emission peak) and exposure research. How does urbanChemFoam capture this transition?*

**Response:** For the nocturnal and daytime transition, a neutral boundary layer is assumed under the idealized setup in this study, although urbanChemFoam itself is capable of modelling effects of thermal stratification through advection and natural convection.

Where (natural) convective effects are to be considered in an infinitely long channel such as a street canyon, for instance, by having the corresponding surfaces maintained at different temperatures, convective cells form in the lengthwise direction of the channel, with characteristic length strongly dependent on the canyon aspect ratio and Rayleigh number. This classical observation has been thoroughly discussed analytically and numerically in Davies-Jones (1970; J Fluid Mech 44/4:695-704) at NCAR, and is termed by the Author as the "finite rolls". In other words, with the exception of a strictly two-dimensional numerical arrangement, the presence of natural convection will render the resulting flow field three-dimensional.

Further, substituting the infinitely long canyon geometry with a periodic finite length canyon, as featured in this study, will result in deviations from this "finite rolls" arrangement. This point has also been asserted by Davies-Jones (1970), and is confirmed, at least empirically, by the Authors' early model runs for the idealized street canyon with convective effects. Hence, under the present idealized street canyon arrangement, not only will the effective two-dimensionality be inapplicable through natural convection, a high degree of grid dependency will also be introduced due to the finite canyon length.

However, it is in the Authors' plan to apply urbanChemFoam on a fully three-dimensional geometry in the future, such as a city block. In this sense, convective effects will be requisite. It is also expected that the ensuing examination / discussion of the nocturnal-to-daytime transition and vice versa will be appreciatively more applicable and interesting.

**Comment:** In addition, radiation effects of buildings, which are interesting on the scale considered, are not included in this study. This could at least be discussed in the form of a possible outlook.

**Response:** The conclusion of the MS has been amended in the paragraph starting with "The results of this study demonstrate …" in Section 5 to indicate the possibility of introducing radiative heating in the future. The amended paragraph reads:

"The results of this study demonstrate the ability for this solver in performing urban-scale chemistry transport modelling at both a stationary and non-stationary capacity in a grid-independent manner. As such, it can be readily adapted to model a larger domain with more realistic geometry, such as a city block. For models covering such domain regions, however, input data such as meteorology and emission profiles need be prescribed at higher spatial and temporal resolutions, which can be accomplished by coupling the urban scale model with outputs from regional models, WRF-Chem for example. Particular emphasis can be placed on the role of absorptive, reflective, and refractive mechanisms of urban structures in the redistribution of solar radiant energy, and their impact on the overall advective and convective thermal stratification in their immediate vicinity."

**Minor Comment 15:** Figs. 3, 5, 8-10: Maybe better include the legend in the figures.

**Response:** After contemplation and experimentation, the Authors have decided against including the key directly on the figure panels to maintain legibility of the contents in the figures in question.

**Minor Comment 16:** Fig. 12: Do the background conditions apply to these results? Some concentrations appear lower than the boundary conditions (cf. Figs. 6 and 12). Suggestion: indicate sun rise/set or night times by grey shadings in the panels.

**Response:** The diurnal profiles of background concentrations have been introduced above the street canyon as non-stationary (time-varying) boundary conditions. The concentrations presented in Figure 12, sampled at various points inside the canyon, are a result of this. Due to mixing and chemical reactions with ground level NO and $NO_2$ emissions, it is expected that the local pollutant concentrations deviate from background levels. The UTC times for sunrise and sunset, defined at the solar zenith transitioning the 90° mark, have been introduced in the MS.

**Response to Reviewer 2**

Thank you very much for the positive comments. To address your remaining concerns, please find our responses following references to your original text below.

> **Comment:** Line 259 – 260: Authors mention that "both the fine and coarse mesh results are in excellent agreement". Can you please provide quantitative data to support this sentence?

**Response:** The Authors have amended the statement in question, to indicate that the agreement between the model results and the LDA measurements for the time-averaged horizontal velocity are inferred from Figure 3.

> **Comment:** Along the manuscript concentrations are provided in ppb, while for regulatory purposes in Europe we work in micro grams per cubic meter. Do you have any specific reason for working in ppb?

**Response:** In the Authors' part, the decision to use molar based concentration – that is, ppb and related units – are largely based on existing regional modelling studies conducted as presented in the MS references (e.g., Kuik et al, 2016, 2017; von Schneidemesser et al, 2017, etc.). However, using ppb, species concentrations are expressed in a way that is independent from molecular mass as well as local thermodynamic state. Therefore, comparison between model results can be done in a standardized fashion, under different ambient and emission conditions, as well as across different mixture constituents.

As a point of interest, species concentrations in OpenFOAM, as well as other open source and commercial computational fluid dynamics (CFD) modelling frameworks, are calculated in terms of mass mixing ratios, that is, the ratio between the mass of the individual species and the mass of the mixture, as indicated in Equation (4) in the MS. Thus conversions to other concentration units relevant for air quality modelling and regulation purposes, such as ppb and micrograms per cubic meter, are necessary and are introduced as a post-processing step.

> **Comment:** Regarding the differences between the simulations in stationary versus transient conditions, do you have any thoughts about which one is the most suited/ accurate? Can you highlight the potential advantages and disadvantages of both conditions?

**Response:** The Reviewer has pointed out one of the major objectives of the study: The comparison between model runs under stationary conditions and non-stationary (transient) conditions. While it is unreasonable to expect stationary model results be applicable over a period of changing ambient conditions, the commonly accepted opinion is that stationary model results would be representative for the point of time at which the ambient conditions are predicated. The development outlined in the present study enabled the examination between stationary and non-stationary model runs.

Two observations can be made from this comparison. First, to the extent of the meteorological conditions considered in this study, there is little change in the in-canyon flow field between stationary and non-stationary runs. This is not surprising, considering that the shear layer in the roof region attenuates the dynamics inside the canyon from those above the buildings. Accordingly, the in-canyon wind should be

less sensitive to the temporal variations in the freestream flow. On the other hand, significant changes can be observed in pollutant concentrations under stationary and non-stationary treatment. As such, temporal variations in emissions and background conditions, which are not considered in stationary runs, are significant and cannot be neglected.

Returning to the Reviewer's question concerning suitability and benefits of each approach, in light of the above observations, where wind and flow distributions are of primary concerns, stationary model runs would likely suffice. On the other hand, if the focus of the study is on pollutant transport and transformation, temporal variations of background concentrations and emissions will have an appreciable influence on model concentrations. As such, a non-stationary approach should be considered.

> **Comment:** The background concentrations are the total concentrations measured at the urban background stations. Is that right?

**Response:** This is correct. The background concentrations presented in Figure 6(C) of the MS are obtained by taking the arithmetic mean over two urban background stations closest to the Silbersteinstraße urban traffic station, where the observation takes place.

---

## Author Response (AR3)

Dear Editor:

In response to your comments made on June 11$^{th}$, 2021, the Authors have made the following amendments outlined below, which can be found in the latest ATC (version 3) file. The Editor's original remarks have been included as reference:

**Editor:** you have answered in detail to the first comment of reviewer 1 regarding the effects on non-neutral stratification, yet none of this discussion ended up in the manuscript, except for a part of a sentence in the outlook. I think it is centrally important to realise for the reader that your setup and tests only apply to neutrally stratified cases. Hence I suggest you add language to the manuscript to address this at an earlier stage (4.1 or 4.2), and discuss the implications along the lines of your reply to reviewer 1.

**Authors:** A brief remarks have been added first in the introduction, and in Section 4.1, to point out that the studies for the idealized street canyon were conducted under neutral conditions.

**Editor:** Regarding the third comment of reviewer 2 on the implications of differences between stationary and transient runs: again, your answer is well taken, but none of it ended up in the manuscript. I concur with reviewer two that it should be more clearly stated what the implications of a transient simulation are, and what you, the authors, suggest to use - and when. Section 5 seems to be a good spot to dwell on this more in detail.

**Authors:** An additional paragraph have been introduced in Section 4.2 to discuss the suitability of stationary and transient model.

Sincerely yours,

The Authors.